# 2D Microporous Covalent Organic Frameworks as Cobalt Nanoparticle Supports for Electrocatalytic Hydrogen Evolution Reaction

**Jialong Song** [1,2,†], **Li Liao** [1,†], **Zerong Zhang** [1], **Yusran Yusran** [1], **Rui Wang** [1], **Jing Fang** [1], **Yaozu Liu** [1], **Yu Hou** [2], **Yujie Wang** [1,*] and **Qianrong Fang** [1,*]

1   State Key Laboratory of Inorganic Synthesis and Preparative Chemistry, Jilin University, Changchun 130012, China; songjl20@mails.jlu.edu.cn (J.S.); liaoli18@mails.jlu.edu.cn (L.L.); zhangzerong@stu.jluzh.edu.cn (Z.Z.); yusran16@mails.jlu.edu.cn (Y.Y.); wangr21@mails.jlu.edu.cn (R.W.); fangjing21@mails.jlu.edu.cn (J.F.); yaozu20@mails.jlu.edu.cn (Y.L.)
2   School of Chemical Engineering and New Energy Materials, Zhuhai College of Science and Technology, Zhuhai 519041, China; houyu@zcst.edu.cn
*   Correspondence: wyujie@jlu.edu.cn (Y.W.); qrfang@jlu.edu.cn (Q.F.)
†   These authors contributed equally to this work.

**Abstract:** Covalent organic frameworks (COFs) are a new class of porous crystalline polymers, which are considered to be excellent supports for metal nanoparticles (MNPs) due to their highly ordered structure, chemical tunability, and porosity. In this work, two novel ultra-microporous COFs, JUC−624 and JUC−625, with narrow pore size distribution have been synthesized and used for the confined growth of ultrafine Co nanoparticles (CoNPs) with high loading. In an alkaline environment, the produced materials were investigated as electrocatalysts for the hydrogen evolution reaction (HER). Electrochemical test results show that CoNPs@COFs have a Tafel slope of 84 mV·dec$^{-1}$, an onset overpotential of 105 mV, and ideal stability. Remarkably, CoNPs@JUC−625 required only 146 mV of overpotential to afford a current density of 10 mA cm$^{-2}$. This research will open up new avenues for making COF-supported ultrafine MNPs with good dispersity and stability for extensive applications.

**Keywords:** covalent organic frameworks; metal nanoparticles; hydrogen evolution reaction

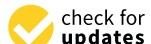



## 1. Introduction

Metal nanoparticles (MNPs), one of the most important catalysts, have gained increasing attention in recent years [1–3]. MNPs often exhibit significant catalytic activity for many reactions under mild conditions. Typically, ultrafine MNPs with smaller sizes demonstrate exceptional catalytic activity. However, due to their high surface energy, ultrafine MNPs tend to aggregate [4,5], with smaller particles showing a stronger aggregation tendency, hence restricting their practical applications, such as catalysts for fuel cell vehicles [6–9]. COFs are a new class of emerging organic porous materials with promising stability and crystallinity that have recently gotten a lot of attention since their first report in 2005 [10–15]. They have demonstrated promising uses in gas storage and separation, heterogeneous catalysis, optoelectronic materials, and energy storage [16–21]. Among various applications, 2D COFs are considered as ideal platforms for MNP supports [22–25]: First of all, COFs have pre-designable well-defined pore structures which can be employed to constrain MNP growth and thus control the size of MNPs; Secondly, the pore channels in COFs are well-isolated, hence minimizing the aggregation of entrapped MNPs [26,27]. Finally, many COFs are stable, which makes them resistant toward decomposition under various reaction conditions. In principle, size-controlled synthesis of ultrafine MNPs could be achieved through rational design of microporous COF templates. However, over the past decade, the majority of published COFs for MNP supports have been mesoporous materials [28–30].

Lately, COF materials with kgd topology have gained considerable attention [31,32]. For example, Zhao [33] and co-workers have developed the first example of 2D kgd-COFs (HAT-NTBA-COF), via Schiff base reaction of HAT-6NH$_2$ and 4,4′,4″-nitrilotribenzaldehyde. Subsequently, Zhang [34] et al. synthesized the ultra-micropores 2D kgd-COFs, via C6-symmetric aldehyde (HFPB) and three different C3-symmetric amines. From the topological point of view, 2D kgd-COFs constructed from C3- and C6-symmetric building units can form the smallest pore size among other two-dimensional topologies. Therefore, they are regarded as good candidates for MNP supports due to their microporous channels. However, 2D kgd-COFs based MNPs supports have never been explored [35–38].

In this work, we report the rational design and synthesis of two microporous 2D kgd-COFs, termed as JUC−624 and JUC−625 (JUC = Jilin University China). Owing to their uniform porosity, good imine Schiff-base complexation ability with Co (II) and high structural stability, small-size Co nanoclusters are uniformly dispersed in 2D layered structured of the COFs, termed as CoNPs@JUC−624 and CoNPs@JUC−625. In alkaline solutions, the catalysts produced show great activity and excellent stability against HER. Especially, CoNPs@JUC−625 requires over potential of only 146 mV to afford a current density of 10 mA cm$^{-2}$ in 1 M KOH alkaline medium. The present results show that electrocatalysts with high crystallinity and robust aromatic frameworks have significantly improved electrocatalytic activity and stability. This study provides a facile strategy to construct microporous 2D kgd-COFs as excellent MNP supports for electrocatalysis.

## 2. Experimental Section

### 2.1. Materials and Instruments

Unless otherwise stated, all starting ingredients and solvents were obtained from J&K Scientific LTD (Beijing, China) and used without further purification. All products were isolated and handled under nitrogen using either glovebox or Schlenk line techniques. The liquid $^1$H NMR spectra were recorded using a AV-400 NMR spectrometer (Bruker BioSpin AG, Fallanden, Swiss). A IRTracer-100 Fourier transform infrared spectrophotometer (SHIMADZU CORPORATION, KYOTO, JAPAN) was used to obtain the FTIR spectra (KBr). Under N$_2$, a DTG-60 thermal analyzer (SHIMADZU CORPORATION, KYOTO, JAPAN) was used to perform a thermogravimetric analysis (TGA). The instrument's operating temperature range was 30 °C to 800 °C, with a heating rate of 10 °C min$^{-1}$ and a N$_2$ flow rate of 30 mL min$^{-1}$. PXRD data were collected on a PANalytical B.V. Empyrean powder diffractometer (PANalytical, ALMELO, Netherlands) using a Cu Kα source (λ = 1.5418 Å) over the range of 2θ = 2.0−40.0° with a step size of 0.02° and 2 s per step. The sorption isotherm for N$_2$ was measured by using a Autosorb-IQ analyzer (Quantachrome, FL, USA) with ultra-high-purity gas (99.999% purity). To estimate pore size distributions for JUC−624 and JUC−625, nonlocal density functional theory (NLDFT) was applied to analyze the N$_2$ isotherm on the basis of the model of N$_2$@77K on carbon with slit pores and the method of non-negative regularization. For scanning electron microscopy (SEM) imaging, MIRA scanning electron microscope (TESCAN, Brno, Czechia) was applied. Electrochemical measurements were carried out using CHI600E (Shanghai CH Instruments Co., Shanghai, China) electrochemical workstation.

### 2.2. Synthesis of JUC−624

TAPB (0.06 mmol, 21.1 mg) and HKH (0.03 mmol, 9.4 mg) were weighted into a Pyrex tube (volume: ca. 20.0 mL with neck length of 9.0 cm and a body length of 18.0 cm), and the mixture was added into 0.5 mL of 1,2-dichlorobenzene, 0.5 mL of 1-Butanol, and 0.1 mL of acetic acid (3 M). The Pyrex tube was flash-frozen in a liquid nitrogen bath, evacuated to an internal pressure of ca. 19.0 mbar and flame-sealed, reducing the overall length by approximately 10.0 cm. After bringing the tube to room temperature, it was placed in a 120 °C oven for 5 days. The resulting precipitate was filtered, exhaustively washed by Soxhlet extractions with dioxane for 48 h. The obtained powder was immersed in anhydrous acetone, and the solvent was exchanged with fresh acetone several times.

The wet sample was then transferred to a supercritical drier (Samdri-PTV-3D), in which the sample was washed six times of liquid $CO_2$, and exchanged with fresh $CO_2$ for six times with the interval of half an hour. The system was heated up to 45 °C to bring about the supercritical state of the $CO_2$, which was bled after half an hour at a very slow flow rate to ambient pressure. The sample was then transferred to a vacuum chamber and evacuated to 20 mTorr under room temperature, yielding brown powder for $N_2$ adsorption measurements.

### 2.3. Synthesis of JUC−625

TAPT (0.06 mmol, 21.1 mg) and HKH (0.03 mmol, 9.4 mg) were weighted into a Pyrex tube (volume: ca. 20.0 mL with a body length of 18.0 cm and neck length of 9.0 cm), and the mixture was added into 0.5 mL of 1,2-dichlorobenzene, 0.5 mL of 1-Butanol and 0.1 mL of acetic acid (6 M). The Pyrex tube was flash-frozen in a liquid nitrogen bath, evacuated to an internal pressure of ca. 19.0 mbar and flame-sealed, reduce the overall length by approximately 10.0 cm. After bringing the tube to room temperature, it was placed in a 120 °C oven for 5 days. The resulting precipitate was filtered, exhaustively washed by Soxhlet extractions with dioxane for 48 h. The obtained powder was immersed in anhydrous acetone, and the solvent was exchanged with fresh acetone for several times. The wet sample was then transfer to a supercritical drier (Samdri-PTV-3D), in which the sample was washed six times of liquid $CO_2$, and exchanged with fresh $CO_2$ for six times with the interval of half hour. The system was heated up to 45 °C to bring about the supercritical state of the $CO_2$, which was bled after half hour at a very slow flow rate to ambient pressure. The sample was then transferred to a vacuum chamber and evacuated to 20 mTorr under room temperature, yielding brown powder for $N_2$ adsorption measurements.

### 2.4. Synthesis of Co(II)NPs@JUC−624 and Co(II)NPs@JUC−625

First, 100.0 mg COF was suspended in 15.0 mL THF and sonicated for 30 min. To this dispersion, 60.0 mg Co(II)-acetate in 2.0 mL $H_2O$ was added. Contents were stirred at RT for 12 h and then the solid was extracted by filtration. The solid was washed with methanol and water. It was then vacuum-dried at 120 degrees Celsius.

### 2.5. Synthesis of CoNPs@JUC−624 and CoNPs@JUC−625

This solid was resuspended in a 3:1 water/methanol mixture and the mixture was heated at 80 °C. Subsequently, the reduction of Co(II) was achieved by addition of 5.0 mL ascorbic acid (1 M). Contents were stirred for 20 h. Finally, the precipitate was filtered and washed with copious amounts of $H_2O$, MeOH, THF and dried under vacuum.

### 2.6. Electrochemical Measurements

Electrochemical test is performed on CHI660E (Shanghai CH Instruments Co., China) electrochemical workstation in a traditional three electrode system. The working electrodes are as-prepared electrodes, the reference electrode is Hg/HgO electrode, and the counter electrode is platinum gauze electrode. The glassy carbon electrode (GCE, $r\frac{1}{4}$ 0.15 mm) is polished with alumina slurry and then washed with ethanol and distilled water. CoNPs@COF electrocatalyst is prepared by mixing 2.0 mg CoNPs@COF electrocatalyst with 1.0 mL ethanol and sonicated and Ketjen Black carbon for 30 min. To prepare electrodes, a drop of the CoNPs@COF electrocatalyst was dropped on GCE and then evaporated in the air. To prevent the catalyst from dissolving in the electrolyte, a small amount of a 0.5% Nafion solution was added to the electrode surface. Every electrochemical testing was conducted in 1 M KOH.

## 3. Results and Discussion

Our strategy for preparing ultra-microporous 2D COFs is based on hexaketocyclohexane knot (HKH, Scheme 1a). As shown in Scheme 1, HKH was designed as a hexagonal 6-connected building unit, and 1,3,5-tris(4-aminophenyl)benzene (TAPB, Scheme 1b) or

2,4,6-tris-(4-aminophenyl)triazine (TAPT, Scheme 1c) was chosen as a typical triangular building unit. The condensation of HKH with TAPB or TAPT produced extended 2D kgd-COFs (Scheme 1f), JUC−624 (Scheme 1d), and JUC−625 (Scheme 1e).

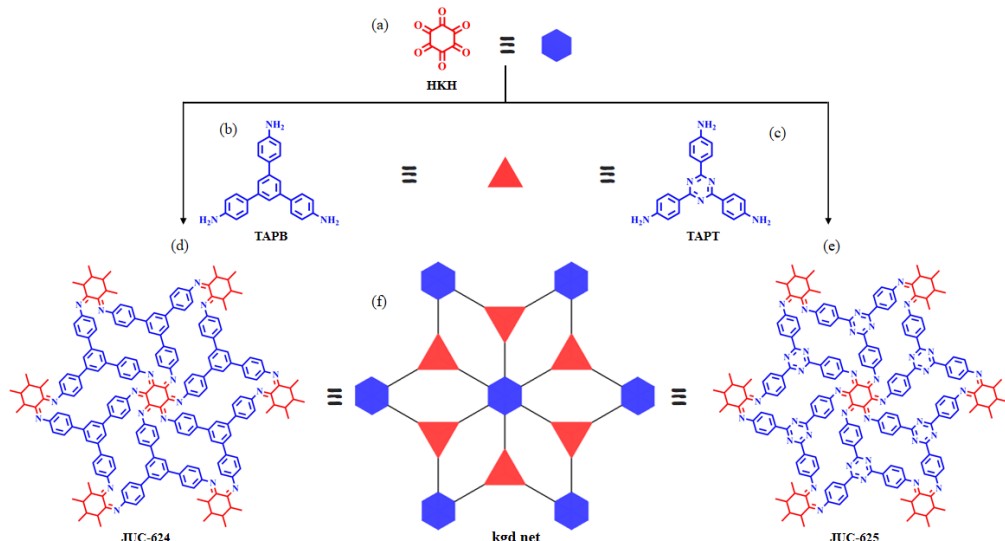

**Scheme 1.** Strategy for preparing 2D kgd-COFs. (**a**) A hexagonal 6-connected building unit, HKH. (**b**,**c**) Two typical triangular building unit, TAPB (**c**) and TAPT (**d**). The condensation of HKH with TAPB or TAPT produced extended 2D JUC−624 (**d**) and JUC−625 (**e**) with a kgd net (**f**).

Typically, the synthesis of the COFs was carried out through the traditional solvothermal approach by suspending HKH and TAPB or TAPT in a mixed solution of 1,2-dichlorobenzene and 1-Butanol containing acetic acid aqueous solution, followed by heating at 120 °C for 5 days. Using PXRD measurements and structural simulations, the crystal structures of JUC−624 and JUC−625 were confirmed (Figure 1). Model structures were obtained through geometrical energy minimization using the Materials Studio software package based on the basis of the AA-stacked kgd net. Both materials adopted space group *P*6 (No. 168) with unit cell parameters of $a = b = 16.2889$ Å, $c = 4.093$ Å, and $\alpha = \beta = 90°$, $\gamma = 120°$ for JUC−624, and $a = b = 16.3501$ Å, $c = 3.915$ Å, $\alpha = \beta = 90°$, $\gamma = 120°$ for JUC−625 (Tables S2 and S3, Supplementary Materials). The calculated PXRD (Figure 1) patterns of the model crystal structures fitted well with the experimental patterns. Furthermore, we carried out full profile pattern matching (Pawley) refinement based on peaks at $2\theta = 6.27, 10.85, 12.55, 16.62, 21.70, 24.33, 25.12,$ and $27.55$ corresponding to the (100), (110), (200), (210), (001), (111), (201), and (320) facets of JUC−624, and at $2\theta = 6.26, 10.85, 12.55, 16.62, 21.72, 24.32, 25.24,$ and $27.55$ corresponding to the (100), (110), (200), (210), (001), (111), (201), and (320) facets of JUC−625, respectively. The refinement results illustrated good agreement factors ($R$p = 2.58% and $R$wp = 3.53% for JUC−624; $R$p = 3.68% and $R$wp = 4.68% for JUC−625). In addition, other alternative structures, such as AB-stacked kgd nets were also considered for both 2D COFs (Figures S14–S19). However, their simulated and experimental PXRD patterns did not match well. Based on the above results, the JUC−COFs are proposed to adopt AA-stacked kgd net architectures, and thus own microporous cavities.

Crystal Morphologies observed under scanning electron microscopy (SEM) analysis revealed the formation of aggerates rod-like NP$_S$ for JUC−624 or needle-like NPs for JUC−625 (Figure 2a,b). Meanwhile, the formation of imine bonds was verified by Fourier transform infrared (FT-IR) spectra with the emergence of new vibration peaks at 1596 and 1604 cm$^{-1}$ for JUC−624 and JUC−625, respectively (Figure S1). High thermal stability up to 400 °C for both COFs was confirmed by thermogravimetric analysis (TGA) under nitrogen atmosphere (Figures S2 and S3). The nitrogen (N$_2$) adsorption isotherm measurements were carried out at 77 K to determine the permanent channel structures and specific surface areas of the COFs. As shown in Figure 2c,d, both JUC−624 and JUC−625 exhibit a

sharp increase gas uptake at low pressure ($p/p_0 < 0.05$), which illustrated the presence of microporous structure. Based on the Brunauer–Emmett–Teller (BET) model, specific surface areas were calculated to be 104 and 134 $m^2 \ g^{-1}$ for JUC−624 and JUC−625, respectively (Figure S4). Pore-size distribution was determined by non-local density functional theory (NL-DFT). JUC−624 showed a narrow pore-size distribution centered at 4.7 Å, while JUC−625 exhibited two kinds of pores of 4.4 Å (Figure S5), which are in good agreement with their proposed models.

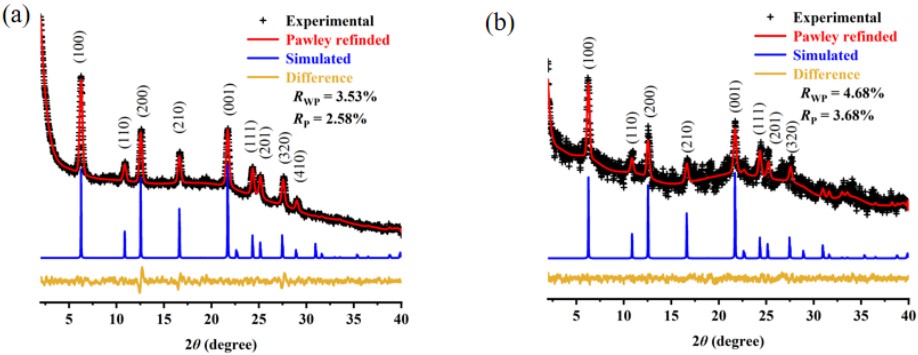

**Figure 1.** PXRD patterns of JUC−624 (**a**) and JUC−625 (**b**).

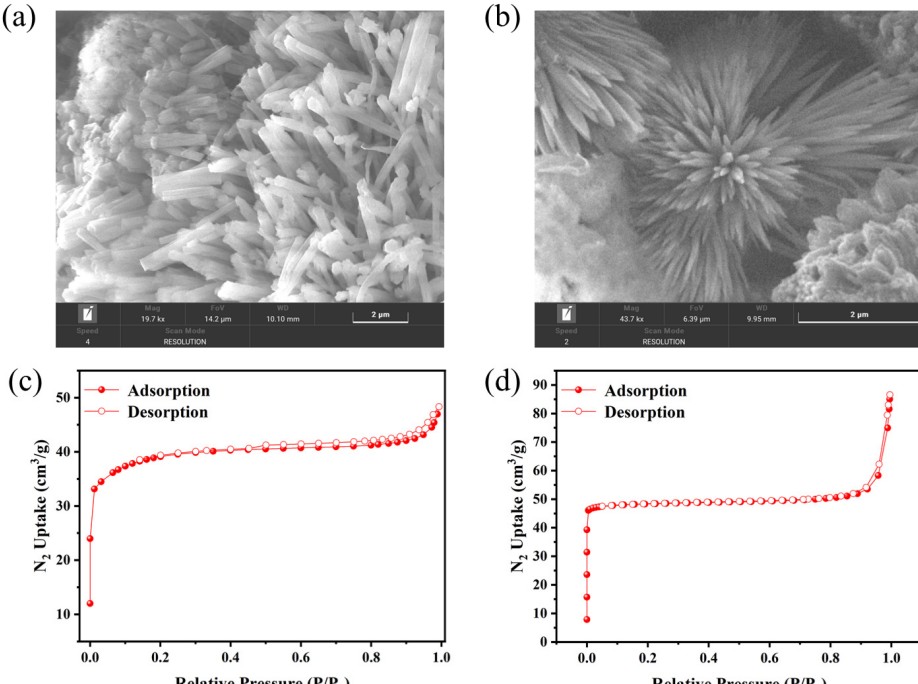

**Figure 2.** SEM images of JUC−624 (**a**) and JUC−625 (**b**). $N_2$ adsorption-desorption isotherms of JUC−624 (**c**) and JUC−625 (**d**).

Given that microporous 2D COFs with well-defined structures are promising templates for the synthesis of MNPs with controllable size and dispersity, we investigated the potential application of utility of both JUC-COFs as templates to guide the nucleation and growth of cobalt (Co) NPs. In a typical procedure, both JUC−624 and JUC−625 were suspended in a methanol solution of Co(II)-acetate to deposit Co ions through metal coordination interactions. Subsequent chemical reduction with ascorbic acid in methanol gave CoNPs@JUC−624 or CoNPs@JUC−625 series. Priorly, before reduction treatment, deposition of Co ions into the COFs was studied by SEM-EDS (energy dispersive spectrum). As shown in Figures S6 and S7, the uniformly dispersed of Co (II) ions are clearly observed,

which is attributed to the good complexation ability of the imine Schiff-base of the COFs with Co(II) ions, as well as the long-range ordered structure of the COFs. Furthermore, elemental analysis by ICP confirmed as high as 2.83 and 2.94 wt% of Co species deposited in the CoNPs@JUC−624 and CoNPs@JUC−625, respectively. These results reveal that the CoNPs were successfully deposited on microporous channels.

To appraise the potential HER catalytic activities of the CoNPs in the JUC-COFs, electrochemical analyses using CoNPs@COFs and unsupported COFs and glassy carbon electrode are performed in alkaline solution (1.0 M KOH, Figures S8–S12). As shown in Figure 3a, over potentials at the current density of 10 mA cm$^{-2}$ are 0.235 and −0.203 mV for JUC−624 and JUC−625, respectively, this shows that the COF alone lacks significant catalytic activity. By contrast, with CoNPs@JUC−625 as electrocatalyst, significantly lower overpotential (146 mV) was required to reach a similar the current density, which is even lower than CoNPs@JUC−624 (176 mV). The outstanding HER catalytic performance of CoNPs@JUC−625 was also confirmed by its small Tafel slope (186 mV·dec$^{-1}$), lower than the value for CoNPs@JUC−624 (190 mV·dec$^{-1}$) (Figure 3c). In addition, the Rct value for CoNPs@JUC−625 was 14.9 Ω, lower than CoNPs@JUC−624 (21.87 Ω), reflecting that CoNPs@JUC−625 possessed the fast charge transfer for HER (Figure 3b). On the other hand, the electrochemical active surface area (ECSA) of the catalyst is directly proportional to the number of active sites, which may be determined by measuring the electrochemical double-layer capacitance (Cdl) at the electrolyte-electrode interface. As can be seen from Figure 3d, CoNPs@JUC−625 possesses Cdl values of 84 mF cm$^{-2}$, which is nearly 2.4 times of JUC−625 (34 mF cm$^{-2}$) or 16.4 times of glassy carbon electrode (5.1 m F cm$^{-2}$). The amperometry i-t Curve test was further carried out to evaluate the stability of CoNPs@COFs. Notably, current densities of −0.92 V can be maintained up to 24 h (Figure S13). All these results confirmed the remarkable electrocatalytic HER performance of CoNPs@JUC−625 in 1 M KOH.

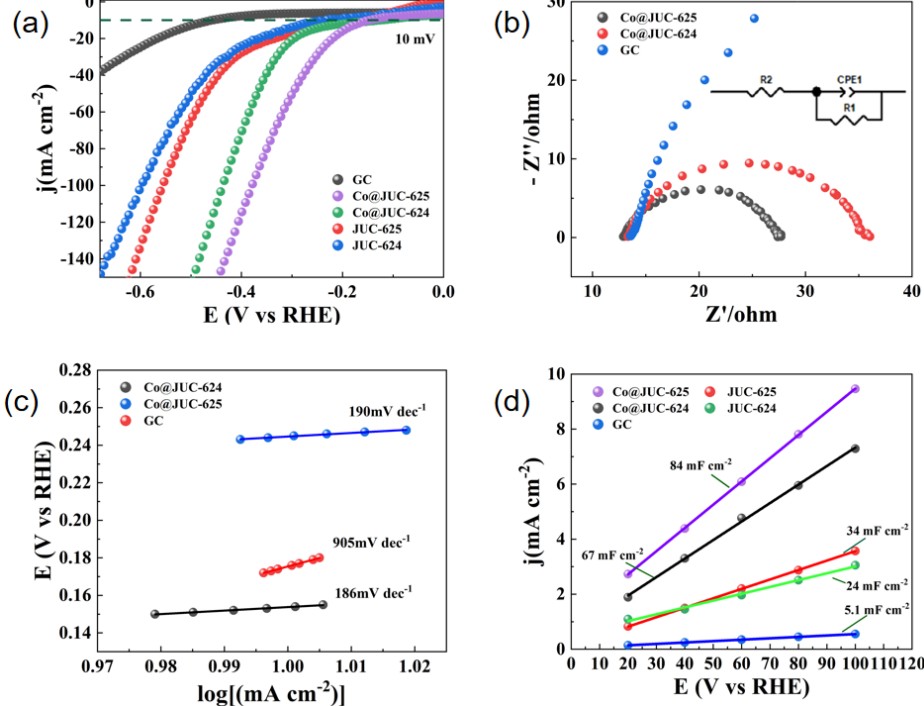

**Figure 3.** (**a**) LSV polarization curves of bare Control, CoNPs@JUC−624, CoNPs@JUC−625, JUC−624 and JUC−625 for HER in 1 M KOH (at a scan rate of 20 mV s$^{-1}$); (**b**) EIS plots of Co NPs@JUC−624, Co NPs@JUC−625 for HER at a catalytically active potential of −0.376 V vs. RHE with inset showing the equivalent circuit; (**c**) corresponding Tafel plots of CoNPs@JUC−624, CoNPs@JUC−625 and bare Control; (**d**) current density as a function of scan rate for bare Control, CoNPs@JUC−624, CoNPs@JUC−625, JUC−624, and JUC−625.

## 4. Conclusions

In summary, we prepared two ultra-microporous 2D kgd-COFs by solvothermal reaction and further uniformly dispersed Co nanoclusters with small particle size. Notably, as an electrocatalyst, CoNPs@JUC−625 only requires an overpotential of 146 mV to afford a current density of 10 mA cm$^{-2}$ in n alkaline solution (1 M KOH). In addition, CoNPs@JUC−625 possesses the highest Cdl (84 mF cm$^{-2}$), which is nearly 2.4 times of JUC−625 (34 mF cm$^{-2}$) and 16.4 times of glassy carbon electrode (5.1 mF cm$^{-2}$), corresponding to the largest ECSA. The remarkable electrocatalytic HER performance of CoNPs@JUC−625 was attributed to the enhanced electrical conductivity and the production of Co nanoparticles that can function as HER active sites. These results indicate that the synthesized CoNPs@JUC−625 can serve as a feasible support for developing Pt-free electrode materials for HER applications.

**Supplementary Materials:** The following supporting information can be downloaded at: https://www.mdpi.com/article/10.3390/cryst12070880/s1, FT-IR spectra, TGA curves, N$_2$ adsorption, SEM mapping, electrochemical HER performance, and unit cell parameters (References [39–43] are cited in the Supplementary Materials).

**Author Contributions:** Conceptualization, J.S., Y.W., and Q.F.; methodology, J.S. and L.L.; software, Z.Z. and Y.H.; validation, L.L., R.W., and Z.Z.; formal analysis, J.S., J.F. and Y.L.; investigation, R.W.; data curation, J.S. and Z.Z.; writing—original draft preparation, J.S., L.L.; writing—review and editing, Y.Y., J.F., and Y.L.; supervision, Y.W. and Q.F.; project administration, Y.W. and Q.F. All authors have read and agreed to the published version of the manuscript.

**Funding:** This research was funded by National Natural Science Foundation of China (22025504, 21621001, 21390394, and 22105082), "111" project (BP0719036 and B17020), China Postdoctoral Science Foundation (2020TQ0118 and 2020M681034), and the program for JLU Science and Technology Innovative Research Team.

**Institutional Review Board Statement:** Not applicable.

**Informed Consent Statement:** Not applicable.

**Data Availability Statement:** Not applicable.

**Conflicts of Interest:** The authors declare no conflict of interest.

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
