# Peer review of "2D Microporous Covalent Organic Frameworks as Cobalt Nanoparticle Supports for Electrocatalytic Hydrogen Evolution Reaction"

_crystals, doi:10.3390/cryst12070880_

Round 1

Reviewer 1 Report

In this work the authors report the successful synthesis of two novel COFs based on hexagonal and tritopic building units. The new COFs were characterized in detail through various techniques such as PXRD, IR-spectroscopy, SEM microscopy, while their porosity has also been studied with the appropriate methods. Furthermore, the COFs that have been prepared were loaded with Cobalt NPs and were further studied as electrocatalysts, especially for Hydrogen Evolution Reaction (HER). The synthetic methods are described in detail as well as the testing of the new COFs for the applications of interest. In my opinion this work can be published in Crystals after the following revisions and suggestions:

  • The COFs that have been prepared in this work are immersed in different solvents during the conduction of the experiments. I believe that the corresponding PXRD diagrams of each COF after its immersion in each solvent (e.g., acetone, toluene, H2O, MeOH and THF) will verify the stability of these COFs throughout the procedure.
  • Also, in the case of the Co-NP loaded COFs, the authors should provide the corresponding PXRD diagrams and discuss the presence (or absence) of the extra peaks that are expected in higher angles due to the presence of metal nanoparticles.
  • A N2 uptake isotherm at 77K of the COFs after the addition of MNPs will provide extra information about the NP-loaded-COFs and their porosity.
  • The text should also be rechecked for minor spelling and typo errors. For example, the word super critical should be updated to supercritical (line 98 and 114), or “was washed with six times” should be updated to “was washed six times” (line 99 and 115).
  • In Figures 2c and 2d the y-axis’ label should be updated to N2 uptake, cm3 (STP) g-1, according to the IUPAC recommendation [ Compendium of Chemical Terminology, 2nd ed. (the "Gold Book"). Compiled by A. D. McNaught and A. Wilkinson. Blackwell Scientific Publications, Oxford (1997). Online version (2019-) created by S. J. Chalk. ISBN 0-9678550-9-8. https://doi.org/10.1351/goldbook.S06036].
  • Regarding to the performance of the reported materials in electrocatalysis applications; I believe that the authors should include an extra table, that will present the comparison between the reported COFs and similar materials and/or representative examples of top performing materials of the field.

Author Response

Reviewer: 1

Recommendation: Publish in Crystals after the following revisions and suggestions:

In this work the authors report the successful synthesis of two novel COFs based on hexagonal and tritopic building units. The new COFs were characterized in detail through various techniques such as PXRD, IR-spectroscopy, SEM microscopy, while their porosity has also been studied with the appropriate methods. Furthermore, the COFs that have been prepared were loaded with Cobalt NPs and were further studied as electrocatalysts, especially for Hydrogen Evolution Reaction (HER). The synthetic methods are described in detail as well as the testing of the new COFs for the applications of interest. In my opinion this work can be published in Crystals after the following revisions and suggestions:

Response: We greatly appreciate the highly positive comments. We have addressed all the reviewer’s suggestions one by one as follows.

[1] The COFs that have been prepared in this work are immersed in different solvents during the conduction of the experiments. I believe that the corresponding PXRD diagrams of each COF after its immersion in each solvent (e.g., acetone, toluene, H2O, MeOH and THF) will verify the stability of these COFs throughout the procedure.

Response: We sincerely thank the reviewer for this question. We have added the stability of these COFs for various solvents in Figure S20.

[2] Also, in the case of the Co-NP loaded COFs, the authors should provide the corresponding PXRD diagrams and discuss the presence (or absence) of the extra peaks that are expected in higher angles due to the presence of metal nanoparticles.

 Response: We thank the reviewer for this question. Co nanoparticles loaded in COF have been proved by ICP. However, no PXRD of Co nanoparticles was observed, which due to amorphous Co nanoparticles or a little percentage of quality.

[3] A N2uptake isotherm at 77K of the COFs after the addition of MNPs will provide extra information about the NP-loaded-COFs and their porosity.

Response: We sincerely thank the reviewer for this question. Due to the pore size of JUC-624 and JUC-625 is the smallest among two-dimensional topologies, NP-loaded-COFs are almost no N2 uptake.

[4] The text should also be rechecked for minor spelling and typo errors. For example, the word super critical should be updated to supercritical (line 98 and 114), or “was washed with six times” should be updated to “was washed six times” (line 99 and 115).

Response: We sincerely thank the reviewer for this careful observation. These problems of spelling and typo errors have been revised.

[5] In Figures 2c and 2d the y-axis’ label should be updated to N2uptake, cm3 (STP) g-1, according to the IUPAC recommendation [ Compendium of Chemical Terminology, 2nd ed. (the "Gold Book"). Compiled by A. D. McNaught and A. Wilkinson. Blackwell Scientific Publications, Oxford (1997). Online version (2019-) created by S. J. Chalk. ISBN 0-9678550-9-8. https://doi.org/10.1351/goldbook.S06036].

Response: We sincerely thank the reviewer for this question. This problem has been revised in Figure 2.

[6] Regarding to the performance of the reported materials in electrocatalysis applications; I believe that the authors should include an extra table, that will present the comparison between the reported COFs and similar materials and/or representative examples of top performing materials of the field.

Response: We sincerely thank the reviewer for this question. The information of reported materials in electrocatalysis applications has been added in Table S1.

Reviewer 2 Report

The paper is well written and deals with a new and important topic, the use of nanomaterials for Hydrogen production.

However, I have some comments:

- Synthesis of JUC-624  and JUC-625:

What happen if the reaction was done using ordinary hydrothermal technique with an autoclave? What is the advantage of using sealed glass tube instead? This need to be mentioned.

- Characterization of CoNPs@JUC-624 and CoNPs@JUC-625:

I recommend doing Xrd for CoNPs@JUC-624 and CoNPs@JUC-625 to confirm the presence of Cobalt nanoparticles in them and whether cobalt species exists as metallic Co(0) or cobalt oxide.

- There is a confusion concerning SEM-EDX, in the text it was mentioned that “Priorly, before reduction treatment, deposition of Co ions into the COFs was studied by SEM-EDS (energy dispersive spectrum). As shown in Figures S6 and S7, the uniformly dispersed of Co (II) ions are clearly observed” This means that SEM-EDX was done for Co(II)NPs@JUC-624 and Co(II)NPs@JUC-625. However, in supplementary the titles of Figures SEM-EDS S6 and S7 refers to CoNPs@JUC-624 and CoNPs@JUC-625. Please revise and correct.

- Electrochemistry, Impedance:

It was mentioned “In addition, the Rct value for CoNPs@JUC-625 was 14.9 Ω, lower than CoNPs@JUC-624 (21.87 Ω), reflecting that CoNPs@JUC-625 possessed the fast charge transfer for HER (Figure 3b).” What is the equivalent electric circuit used in fitting the electrochemical data to calculate Rct. It is not mentioned. Please mention the name of equivalent electric circuit, and put it as drawing inside of Figure 3b.

Author Response

Recommendation: Publish in Crystals after the following revisions and suggestions:

Reviewer's comments: The paper is well written and deals with a new and important topic, the use of nanomaterials for Hydrogen production. However, I have some comments.

Response: We greatly appreciate the positive comments on our results. We have addressed all the reviewer’s suggestions one by one as follows.

 [1] What happen if the reaction was done using ordinary hydrothermal technique with an autoclave? What is the advantage of using sealed glass tube instead? This need to be mentioned.

Response: We thank the reviewer for the question. This material can be also synthesized in the autoclave. However, getting a small amount of product is more suitable for the use of sealed glass tubes.

 [2] I recommend doing Xrd for CoNPs@JUC-624 and CoNPs@JUC-625 to confirm the presence of Cobalt nanoparticles in them and whether cobalt species exists as metallic Co(0) or cobalt oxide.

Response: We sincerely thank the reviewer for this suggestion. Co nanoparticles loaded in COF have been proved by ICP. However, no PXRD of Co nanoparticles was observed, which is due to amorphous Co nanoparticles or a little percentage of quality.

[3] There is a confusion concerning SEM-EDX, in the text it was mentioned that “Priorly, before reduction treatment, deposition of Co ions into the COFs was studied by SEM-EDS (energy dispersive spectrum). As shown in Figures S6 and S7, the uniformly dispersed of Co (II) ions are clearly observed” This means that SEM-EDX was done for Co(II)NPs@JUC-624 and Co(II)NPs@JUC-625. However, in supplementary the titles of Figures SEM-EDS S6 and S7 refers to CoNPs@JUC-624 and CoNPs@JUC-625. Please revise and correct.

Response: We thank the reviewer for this careful observation. We have revised and corrected this mistake.

[4] It was mentioned “In addition, the Rct value for CoNPs@JUC-625 was 14.9 Ω, lower than CoNPs@JUC-624 (21.87 Ω), reflecting that CoNPs@JUC-625 possessed the fast charge transfer for HER (Figure 3b).” What is the equivalent electric circuit used in fitting the electrochemical data to calculate Rct. It is not mentioned. Please mention the name of equivalent electric circuit, and put it as drawing inside of Figure 3b.

Response: We thank the reviewer for the suggestion. This equivalent electric circuit has been provided in Figure 3b.

Reviewer 3 Report

Detailed comments:

1.      The English of the text should be checked

2.      The novelty of manuscript should be highlighted more

3.      Please eliminate multiple references. After that, please check the manuscript thoroughly and eliminate ALL the lumps in the manuscript. This should be done by characterizing each reference individually and by mentioning 1 or 2 phrases per reference to show how it is different from the others and why it deserves mentioning. Multiple references are of no use for a reader and can substitute even a kind of plagiarism, as sometimes authors are using them without proper studies of all references used. In the case, each reference should be justified by it is used and at least short assessment provided.

4.      Lines 30-31, authors write: “As a result, their gradually catalytic activity loses 30 over time, and thus limiting their practical applications” – the practical applications must be indicated

5.      For parts 2.2, 2.3, 2.4 or 2.5 – the schematically diagrams must be included, one or more, for more clarity

6.      For 2.6 Electrochemical measurements – a picture or a photo must be included

7.      Comparison between the obtained results and measured in this study with other reported studies should be done and included for more clarity (indicate values not just number of reference).

8.      The possible applications of the obtained materials must be included

Author Response

Recommendation: Publish in Crystals minor revisions.

Response: We greatly appreciate the positive comments on our results. We have addressed all the reviewer’s suggestions one by one as follows.

[1] The English of the text should be checked.

Response: We thank the reviewer for this suggestion. The English of the text has been double-checked.

[2] The novelty of manuscript should be highlighted more.

Response: We sincerely thank the reviewer for the suggestion. The novelty of manuscript has been highlighted more in the revised manuscript.

[3] Please eliminate multiple references. After that, please check the manuscript thoroughly and eliminate ALL the lumps in the manuscript. This should be done by characterizing each reference individually and by mentioning 1 or 2 phrases per reference to show how it is different from the others and why it deserves mentioning. Multiple references are of no use for a reader and can substitute even a kind of plagiarism, as sometimes authors are using them without proper studies of all references used. In the case, each reference should be justified by it is used and at least short assessment provided.

Response: We thank the reviewer for these questions. We have eliminated multiple references and the lumps in the manuscript.

[4] Lines 30-31, authors write: “As a result, their gradually catalytic activity loses 30 over time, and thus limiting their practical applications” – the practical applications must be indicated.

Response: We thank the reviewer for this careful observation. We have corrected this part.

[5] For parts 2.2, 2.3, 2.4 or 2.5 – the schematically diagrams must be included, one or more, for more clarity.

Response: We thank the reviewer for these questions. We have added the schematic diagrams in Scheme S1, Scheme S2 and Scheme S3.

[6] For 2.6 Electrochemical measurements – a picture or a photo must be included.

Response: We thank the reviewer for this question. We have added the electrochemical measurements in Scheme S4.

[7] Comparison between the obtained results and measured in this study with other reported studies should be done and included for more clarity (indicate values not just number of reference).

Response: We sincerely thank the reviewer for this question. The information of reported materials in electrocatalysis applications has been added in Table S1.

[8] The possible applications of the obtained materials must be included.

Response: We thank the reviewer for this question. We have investigated electrocatalytic hydrogen evolution of the obtained materials.

Round 2

Reviewer 1 Report

The authors have successfully replied to the comments that have been done earlier. However, I would suggest a detailed revision regarding typos and spelling mistakes, before publishing,  as it would improve the overall presentation of their work.   

Reviewer 2 Report

The paper can be accepted